# Anastomotic Leakage after Colorectal Surgery in Ovarian Cancer: Drainage, Stoma Utility and Risk Factors

**DOI:** 10.3390/cancers14246243

**Published:** 2022-12-18

**Authors:** Liliana Mereu, Francesca Dalprà, Valeria Berlanda, Riccardo Pertile, Daniela Coser, Basilio Pecorino, Maria Gabriella D’Agate, Francesco Ciarleglio, Alberto Brolese, Saverio Tateo

**Affiliations:** 1Azienda Provinicale Servizi Sanitari, 38123 Trento, Italy; 2Gynecologic and Obstetric Department, Ospedale Cannizzaro, Catania and Kore University, 94100 Enna, Italy; 3Gynecologic and Obstetric Department, Ospedale Santorso, 36014 Vicenza, Italy; 4Gynecologic and Obstetric Department, Santa Chiara Hospital, APSS Trento, 38123 Trento, Italy; 5Service of Clinical and Evaluative Epidemiology, APSS Trento, 38123 Trento, Italy; 6Radiology Department, Santa Chiara Hospital, APSS Trento, 38123 Trento, Italy; 7Department of General Surgery, Valli del Sole Hospital, APSS Trento, 38123 Trento, Italy; 8Department of General Surgery & HPB Unit, Santa Chiara Hospital, APSS Trento, 38123 Trento, Italy; 9Service Gynecologie Obstetrique, Centre Hospitalier de Troyes CHT, 10000 Troyes, France

**Keywords:** ovarian cancer, debulking surgery, anastomosis leakage, abdominal drainage, stoma, rectosigmoid resection

## Abstract

**Simple Summary:**

Anastomosis leakage is a serious postoperative complication after colorectal resection for ovarian cancer that can lead the delay of first line chemotherapy. Known risk factors for anastomosis leakage are age, Charlson Comorbidity Index, serum albumin level, prior chemotherapy or radiotherapy, number and length of bowel resection, level of anastomosis close to anal verge, absence of protective stoma. Intrabdominal drains and protective stoma may be used only in selected cancer ovarian patients undergoing debunking surgery with rectosigmoid resection.

**Abstract:**

Objective: to evaluate the incidence of anastomotic leakage (AL), risk factors and utility of drainage and stoma in patients undergoing intestinal surgery for ovarian cancer in a single institution and in a review of the literature. Methods: retrospective study that includes consecutive patients undergoing debulking surgery with en bloc pelvic resection with rectosigmoid colectomy for ovarian cancer between 1 November 2011 and 31 December 2021. Data regarding patient and tumour characteristics, surgical procedure, hospitalisation, complications and follow-up were recorded and analysed. The PubMed database was explored for recent publications on this topic. Results: Seventy-five patients were enrolled in the study. All anastomoses were performed at a distance of >6 cm from the anal margin, with negative leak tests and tension-free anastomosis. Diverting stoma were performed in just three patients (4%). At least one perianastomotic pelvic drain was positioned in 71 patients (94.7%) and was removed on average on postoperative day 7. Four patients (5.3%) experienced AL. In all cases, the drain content was not the only sign of complication, as the clinical signs were also highly suggestive. Just one patient received conservative treatment. Average postoperative hospitalisation was 14.6 days (SD: ±9.7). There were no deaths at 30 and 60 days after surgery. Between the AL and non-AL groups, statistically significant differences were observed for age, Charlson Comorbidity Index, length of the intestinal resection and fitness for chemotherapy at 30 days. In ovarian cancer, rectosigmoid resection is a standardised procedure with comparable results for AL, and risk factors for AL are discretely homogeneous. What is neither homogeneous nor standardised according to the literature is the use of stomas and/or drains. Conclusion: use in the future of protective stoma and/or intra-abdominal drains is to be explored in selected and standardised situations to verify their preventive role.

## 1. Introduction

Epithelial ovarian cancer is the primary cause of death from gynaecological tumours amongst women in industrialised countries. Most cases are diagnosed at an advanced stage of disease, and post-surgical residual disease is a predictive factor for survival [1]. Ovarian cancer primarily spreads within the peritoneal serous membranes and frequently involves the recto-uterine pouch, sigmoid colon and rectum. Optimal debulking is obtained by means of a surgical procedure in which all visible disease is removed, which often involves one or more bowel resections, most commonly of the sigmoid colon and rectum with an en bloc pelvic resection [2]. The most serious complications of this surgical procedure include anastomotic leakage, which usually requires the patient to have another surgery and causes considerable postoperative morbidity and mortality. A number of retrospective studies have evaluated the various factors that are closely related to an increased incidence of this complication [3,4,5,6]. The therapy for advanced ovarian cancer involves subsequent chemotherapy treatment, and anastomotic leakage can lead to delays in starting first-line chemotherapy and, in some cases, may even make it impossible.

Intraoperative manoeuvres that have been studied to reduce AL incidence are: mechanical anastomosis, avoiding IMA transection, intrabdominal drains placement and diverting stoma. The placement of intra-abdominal drains, which was proposed in cases of bowel surgery by Billroth and Sims in the late 1800s, has always been considered useful both for avoiding anastomotic leaks by preventing the accumulation of blood and serum, potential vectors of infection, and for allowing the identification of an anastomotic leak through the observation of the drainage tube contents [7]. However, a number of studies have reported drainage-related complications, including infections, pressure sores, perianastomotic fluid collections due to incorrect placement, formation of adhesions due to the inflammatory state and intestinal occlusions [8,9]. The presence of an intra-abdominal drain prolongs the duration of hospitalisation. Routine drain placement following bowel resection would not appear to be justified considering the associated complications: infections, abscesses and fistulae [10,11,12]. Despite the presence of data that do not show an advantage for abdominal drain placement, this procedure is still common practice during surgical procedures.

A protective diverting stoma for the prevention of anastomotic leakage may have a role to play in the highest-risk cases [13]: reduced distance from the anus (<6 cm), formation of abscesses or significant contamination with stools, poor-quality anastomosis, positive leakage test, non-tension-free anastomosis, impaired tissue quality and manual anastomosis.

The main objective of the study was to evaluate, in our experience, the relevance of anastomotic leakage in patients undergoing intestinal surgery for ovarian cancer in relation to protective stoma and intra-abdominal drainage with a literature review. The secondary objective was to identify a population with a higher risk of developing anastomotic leakage and that would therefore benefit from the prophylactic placement of an intra-abdominal drain, protective ileostomy, colostomy (diverting ileostomy or colostomy) or ghost ileostomy.

## 2. Materials and Methods

The sample of this retrospective study includes consecutive patients undergoing debulking surgery with en bloc pelvic resection with rectosigmoid colectomy for ovarian cancer (primary surgery, interval surgery or surgery for recurrence) at the Gynaecologic Oncology Unit of the Ospedale Santa Chiara in Trento between 1 November 2011 and 31 December 2021. The study was approved by the ethics committee of the APSS (Provincial Health Authority).

The surgical procedures were performed by the same gynaecological oncological surgeon using a standard procedure [14], with the assistance, for the intestinal anastomosis, of colleagues from the General Surgery Unit. In all cases, no mechanical intestinal preparation was provided, and intravenous antibiotic prophylaxis was administered.

The inclusion criteria were: histological diagnosis of ovarian/Fallopian tube/peritoneal neoplasm at the definitive exam and rectosigmoid tract resection with a cytoreductive intent.

Data regarding each patient’s medical history and characteristics, the stage and characteristics of the tumour, the surgical procedure and clinical evolution during hospitalisation, complications and subsequent disease follow-up were acquired by consulting the HIS (Hospital Information System), OncoSys (Oncological System) and medical records, and were entered in a common database created specifically for the purpose of this study.

For all patients, we analysed the procedures performed and the post-surgical complications classified using the Clavien–Dindo system [15].

The disease stage was determined using the FIGO 2014 staging system [16].

Patients’ general health was analysed using two validated scoring systems: the ASA (American Society of Anaesthesiologists) physical status classification system [17] and the ECOG (Eastern Cooperative Oncology Group) performance status [18], along with the use of the age-adjusted Charlson Comorbidity Index (CCI) [19]. We evaluated the potential risk factors for anastomotic fistula: patient-related factors (general health, alcohol consumption, diabetes mellitus, steroid therapy, prior abdominal surgery, prior pelvic radiotherapy, treatment with bevacizumab and serum albumin values); tumour-related factors (presence of ascites, histology, stage, grading and surgical indication for rectosigmoid resection surgery in terms of the surgical infiltration depth); intraoperative factors (operating time, residual tumour, blood loss and intraoperative transfusions and body temperature at the end of surgery); and resection- and anastomosis-related factors (length of resection, type of anastomosis and procedure used and other associated bowel resections).

The authors recorded postoperative patient management data (introduction of solids feeding, duration of hospitalisation, mobilisation, bladder catheter removal and intra-abdominal drain removal), fitness for chemotherapy 30 days after surgery and mortality at 60 and 90 days after surgery.

A subsequent analysis was performed to split the sample into two groups based on the complication associated with the greatest morbidity, anastomotic leak. Two sub-groups were therefore created: one consisting of patients with the postoperative complication anastomotic leak and one consisting of patients who did not receive this diagnosis.

### 2.1. Statistical Analysis

Data are expressed as means and standard deviations in cases of quantitative variables, and as tables of observed frequencies and percentages in cases of categorical variables. On the basis of the analysed variables and their possible normal distribution (for quantitative variables only), different statistical tests were applied: Fisher’s exact test, Student’s *t*-test (with equal or unequal variances) and the Kruskal–Wallis test. In addition, univariate logistic regression analyses were performed on the probability of the presence of fistula (with OR and 95% CI). For each test, statistical significance was achieved with a *p*-value ≤ 0.05.

Statistical data were analysed using the Microsoft Excel^®^ and SAS-system^®^ software (version 9.4).

### 2.2. Literature Review

Studies were selected by exploring the PubMed database with a combination of the keywords “ovarian cancer”, “colorectal surgery” and “leakage anastomosis”. We took into consideration studies published after the year 2000.

## 3. Results

Seventy-five consecutive patients who underwent rectosigmoid resection during surgery for ovarian cancer were enrolled in the study. Patient and tumour characteristics are described in Table 1 and Table 2, respectively.

Regarding the data associated with surgery (Table 3), in almost all cases (74, 98.7%) the surgical procedures were performed on an elective basis; in just one (1.3%), the patient had already started the diagnostic work-up for a bowel obstruction and suspicious ovarian cancer.

The mean reported operating time was 335.7 min (±79.8), which confirmed the complexity of the clinical setting. For 68 women (89.3%), surgery made it possible to obtain optimum debulking (no residual tumour). The presence of miliary carcinomatosis at the end of the surgical procedure was reported in just one patient.

In 21% of cases intra- or postoperative red blood cell transfusions were required, and the mean blood loss was 415.4 mL (±380.1). Between induction of anaesthesia and waking, patients were kept in conditions of “normothermia” with the aid of dedicated equipment when necessary: the mean body temperature at the end of the surgical procedure was 36.3 °C (±0.6 °C).

At the final histology exam, intestinal wall involvement was superficial, with infiltration of the serosa in 40 patients (53%), the muscularis in 18 (24%), the submucosa in 14 (18.7%) and the mucosa in 3 (4%).

All anastomoses were performed at a distance of >6 cm from the anal margin using mechanical stapler, except one, which was performed manually. For all the patients in the surgical register, leak tests (via methylene blue test) were negative for leakage (although in one case, 1.3%, the washers were seen not to be intact), and the anastomosis was tension-free.

At least one perianastomotic pelvic drain was positioned in 71 patients (94.7%) and was removed on average on postoperative day 7. In most cases a tube drain was used (42 patients, 59.1%), and in the remaining cases a Jackson–Pratt drain was used (29, 40.9%). In 31 patients (41.3%) two drains were placed; in 14 patients they were perianastomotic (45.1%), and in 10 (32.3) they were pelvic. On average, the second drain was removed on postoperative day 8; 24 (77.4%) were tube drains, and 7 (22.6%) were Jackson–Pratt drains. No drain was placed in just four patients (3%).

In three cases (3/75, 4%) a protective diverting stoma was performed for abscess with intestinal perforation, multiple bowel resections and impaired tissue quality.

A total of 49 patients (65.3%) required postoperative monitoring with admission to the postoperative ICU for more than 6 h.

On average, solids feeding was resumed on day 4 or day 5 (4.5 ± 0.71 days). Further nutritional support was required for 24 patients (33%) for an average of almost 8 days (7.91 ± 3.81 days).

Complications were detected following Clavien–Dindo classification, and the reported complication occurrences among the 75 patients are presented in Table 4a; patient-related complications are presented in Table 4b.

For all four patients (5.3%) who experienced anastomotic fistula, the drain content was not the only sign of the complication, as the clinical signs were also highly suggestive. All cases with clinical suspicion of leakage underwent imaging investigations (CT scan of the chest and abdomen) and were examined by the general surgeon in order to define the diagnostic and therapeutic programme. Just one patient received conservative treatment, and the others underwent further surgery with diverting stoma.

The average postoperative hospitalisation was 14.6 days (SD: ±9.7). There were no deaths at 30 and 60 days after surgery. One patient died within 90 days of the surgical procedure.

Seventy-one patients (94.7%) received chemotherapy after surgery; it was not possible to continue the treatment programme in just four patients one in the AL group and three in the no-AL group.

Overall, 82.7% (62) of patients were fit to start chemotherapy within 30 days of surgery.

In order to achieve the objective of the study, a statistical analysis was performed considering the population that developed the complication anastomotic leakage (four patients, 5.3%).

All patients with AL had a drain but no prophylactic intraoperative stoma positioning during primary surgery.

As far as the risk factors (Table 5) are concerned, statistically significant differences were observed for age (OR: 1.15, 95% CI: 0.98–1.34) and the Charlson Comorbidity Index (OR: 3.46, 95% CI:1.26–9.49) alone.

Regarding the tumour characteristics and intraoperative variables, the statistical analysis revealed a statistically significant difference in the length of the intestinal resection (Table 6). No statistically significant differences were observed for all the other parameters analysed, most notably the technique used to perform the anastomosis.

No statistically significant differences between the two study populations were observed with regard to the location, type and number of intra-abdominal drains.

A statistically significant difference was, however, observed with regard to fitness for chemotherapy at 30 days: 85.9% vs. 25% *p* = 0.0152. All the patients with AL started chemotherapy treatment.

The results of the literature review are summarized in Table 7.

## 4. Discussion

Leakage, especially colorectal leakage, is the most dreaded complication following intestinal surgery. The incidence rate reported for anastomotic leakage following resection for bowel cancer varies from 1.2 to 15% [32]; whereas for ovarian cancer, the incidence rate varies from 0.8 to 6.8% [3,4,6].

The incidence data for anastomotic leakage observed in this study are in line with those reported in other studies, with a rate of 5.3%, although the protective stoma rate (4%) was lower than those (21.6–58.5%) reported in the literature of ovarian cancer, as shown in Table 7. These data confirm that the high standardisation of the pelvic debulking surgical procedure can yield similar results in facilities managing different volumes but that meet the quality criteria [33].

The literature identifies a number of risk factors for this complication [5,24], including factors intrinsic to the patient, factors associated with the type of procedure and factors associated with the type of resection and anastomosis. As far as the factors associated with patient characteristics are concerned, in our caseload, statistically significant differences were observed for age and the Charlson Comorbidity Index. In this study, patients with an anastomotic fistula had a significantly higher Charlson Index than the group without a fistula, and a 1-point increase in the Charlson Comorbidity Index (CCI) correlated with a 3.5% increase in the likelihood of fistula, whereas the increase with age was just 1.1%. ASA score and PS, on the other hand, do not appear to influence the appearance of this complication. These data confirm that patient morbidities, particularly the CCI, can be considered an important risk factor for anastomotic leakage and should be taken into account when performing colorectal surgery [34].

In the literature, age is still a debated factor for AL. Pirrera et al. in their work do not consider age even greater than 80 as prognostic factor of a worse postoperative outcome [35].

We have to underline that even if we found a possible relation between age and AL, our analysis could be weakened by the low number of complicated cases.

As it regards the risk factors associated with the type of surgery, literature data [6] report that the risk factors include a non-tension-free anastomosis at the end of the procedure, a manual technique, non-intact washers and intraoperative leak tests positive for leakage. In the present study, all the resections were intraperitoneal and mechanically performed; our data make it possible to identify a statistically significant difference for the length of the resection, a factor that is more objective than the “tension-free” variable, and which was greater among patients who developed anastomotic leakage, probably due to anastomotic tension or devascularization.

Regarding intraoperative preventive manoeuvres, in colorectal surgery, diverting ileostomy has been proposed as a protective method to reduce the incidence and consequences of AL, although this procedure does not seem to reduce the leak rate, but only related complications such as sepsis, re-laparotomy, etc., and it could be related to many other complications associated with dehydration, malnutrition, renal failure, prolapse and stenosis of the stoma [36]. Reviewing studies on bowel resection in ovarian cancer patients (Table 7), the rate of protective stoma varies from 0 to 53.8%. Only one study reported a role of protective stoma in reducing AL [5].

Stoma does not reduce the incidence of anastomotic leakage, which is more related to vascularization and anastomotic technique, but makes the complication less serious due to peritonitis prevention. In the present study, stoma were performed in just three patients (4%) who were considered to have a high risk of fistula due to an intraoperative finding. Using diverting stoma only in selected patients seems to reduce the incidence and consequent complications reported in other studies on similar populations [6,13,36] without a relevant increase of AL complication.

Prophylactic placement of a pelvic drain remains a common practice, although various studies have published contradictory results in terms of the advantages and disadvantages. The majority of data originates from studies on colorectal surgery for primary lesions in the large intestine [8,9,37], which are undoubtedly more numerous than those regarding patients with ovarian cancer [4,22]. The adoption of ERAS protocols also in the gynaecological surgery field is further calling into question the indiscriminate use of prophylactic intra-abdominal drains [21].

A meta-analysis published in 2019 by Podda et al. [32], analysing the data of four randomised studies on drain placement vs. no drain after colorectal resection, in which no distinction is made regarding the level of the anastomosis (intraperitoneal or sub-peritoneal) suggests that there are no advantages for the prophylactic placement of an intra-abdominal drain in terms of diagnosis of anastomotic leakage, morbidity and mortality, wound infection, incidence of pelvic peritonitis and sepsis, respiratory complications and duration of hospitalisation.

The colorectal literature also includes studies that report an advantage for rectal resections with a distance of <6 cm from the anal margin, which appear to benefit from placement of a drain [7,13,38,39]: in these cases, early diagnosis of anastomotic leakage may be useful [40] also in reducing the morbidities associated with this complication [41].

According to published data, drains contribute to the early diagnosis of anastomotic leakage with a very variable sensitivity of between 5% and 71.4% [7,42]; however, they are not the only alarm signal: the other symptoms reported include triad fever, leucocytosis and pelvic or perineal pain [43] and inexplicable fever associated with tachycardia [44]. They can also be used for monitoring purposes in the case of conservative treatment. In the four patients who developed anastomotic leakage, the other findings that suggested fistulisation included fever, inflammatory marker elevation, asthenia, hypotension, tachycardia and the need for supportive oxygen therapy. In all cases, an emergency CT scan of the abdomen was performed in order to confirm the clinical suspicion. All these patients presented the risk factors identified in this study: old age, higher Charlson Index and longer intestinal tract resection. In our sample, all patients with anastomotic leakage had a drain (and two patients had two drains at the time of diagnosis). It can therefore be concluded that, in the population analysed in this study, the drain was not a decisive element for diagnosis, as the concomitant clinical signs were suggestive in all four cases; in one case (25%) the drain proved to be useful for monitoring the evolution of the conservative treatment chosen. Conservative treatment may sometimes be realised with antibiotic treatment and percutaneous drainage [22,45].

Published studies indicate anastomotic fistula as being a complication associated with considerable morbidity, with a delay in the subsequent chemotherapy treatment [36]. Our data confirm this finding; indeed, a statistically significant difference was observed regarding the possibility of starting chemotherapy at 30 days from the surgical procedure in patients diagnosed with anastomotic leakage.

Despite the limitations associated with the retrospective nature of the study and the absence of a counterpart, the data obtained in this study show that preventive intraoperative manoeuvres such as diverting ileostomy and intrabdominal drains are possibly best used no longer as routine practices, but rather only employed in a selected population identified on the basis of an analysis of the risk factors and intraoperative variables. The analysis of our results constituted a preliminary step for participation in prospective randomised studies that may support these indications.

## 5. Conclusions

Intrabdominal drains and protective stoma may be used only in selected cancer ovarian patients undergoing debunking surgery with rectosigmoid resection.

## Figures and Tables

**Table 1 cancers-14-06243-t001:** Patient characteristics.

	Patients n. 75
Age in years, mean ± SD *	61.7 ± 11.4
BMI ^§^ in kg/m^2^, mean ± SD *	25.01 ± 4
ASA^¶^, n (%)	
1	10 (13.3)
2	58 (77.3)
3	7 (9.4)
Charlson Index, mean ± SD *	7.55 ±1.6
ECOG ** Performance status n (%)	
0	37 (50.7)
1	38 (49.3)
Alcohol n (%)	2 (2.7)
Smokers n (%)	8 (9.7)
Diabetes n (%)	3 (4)
Steroid therapy n (%)	1 (1.3)
Prior abdominal surgery n (%)	54 (72)
Prior pelvic radiotherapy n (%)	1 (1.3)
Prior bevacizumab, n (%)	2 (2.7)
Preoperative albumin gr/dL, mean ± SD *	33.2 ± 6.6

* SD: standard deviation; ^§^ BMI: body mass index; ASA^¶^ score: American Society of Anaesthesiologists score; ECOG ^**^ performance status: Eastern Cooperative Oncology Group performance status.

**Table 2 cancers-14-06243-t002:** Tumour characteristics.

		Patients n. 75
Ascites n (%)	0 <500 mL 500–1500 mL 1500–3000 mL >3000 mL	36 (48)
		15 (20)
		3 (4)
		10 (13.3)
		11 (14.7)
Histology n (%)	○ Serous ○ Mucinous ○ Clear-cell ○ Endometrioid ○ Non-epithelial	66 (87.8)
		1 (1.3)
		2 (2.6)
		5 (7)
		1 (1.3)
Grading n (%)	1 2 3 N/A	0 (0)
		5 (7.1)
		65 (92.9)
		0 (0)
Stage n (%)	○ I ○ II ○ A ○ B ○ III ○ A ○ B ○ C ○ IV ○ A ○ B	0 (0) 0 (0) 2 (2.7) 4 (5.3) 4 (5.3) 48 (64) 1 (1.3) 16 (21.4)

**Table 3 cancers-14-06243-t003:** Surgical characteristics.

Procedures performed n(%) ○ Hystero-oophorectomy ○ Appendectomy ○ Omentectomy ○ Lymphadenectomy ○ Diaphragmatic resection ○ Splenectomy ○ Removal of periportal hepatic lymph nodes ○ Liver resection ○ Bladder resection/nephrectomy/ureteral resection ○ Ileal resection ○ Transverse colon resection ○ Caecal resection ○ Ascending colon resection	64 (85.3) 21 (28) 61 (81.3) 46 (61.3) 23 (30.7) 10 (13.7) 0 (0) 1 (1.3) 2 (2.7) 7 (9.3) 2 (2.7) 2 (2.7) 1 (1.3)
Resection in cm, mean ± SD *	17.9 ± 7.1
Distance form anal verge n (%) ○ >6 cm ○ <6 cm	75 (100) 0
Type of anastomosis n (%) ○ End-to-end ○ End-to-side ○ Pouch ○ Side-to-side	62 (82.7%) 11 (14.6) 0 (0) 2 (2.7)
Anastomosis procedure n (%) ○ Mechanical stapler ○ Mechanical stapler + reinforcing sutures ○ Manual sutures	59 (78.7) 15 (20) 1 (1.3)
Surgical indication n (%) ○ Primary surgery ○ Interval surgery ○ Recurrence surgery ○ Urgent	62 (82.7%) 4 (5.3) 8 (10.7) 1 (1.3)
Operating time in minutes, mean ± SD *	335.7 ± 79.8
Residual tumour n (%) ○ Absent ○ <1 cm ○ >1 cm ○ Carcinomatosis	51 (68) 17 (22.7) 6 (8) 1 (1.3)
Intraoperative blood loss, mL mean ± SD *	415.4 ± 380.1
Intra/postoperative transfusions n (%) ○ None ○ RBC ** ○ Plasma ○ Platelets	57 (76) 16 (21.3) 2 (2.7) 0 (0)
Body temperature at end of procedure °C, mean ± SD	36.3 ± 0.6

* SD: standard deviation; ** RBC: Red Blood Cell.

**Table 4 cancers-14-06243-t004:** Complication occurrences.

**a. Postoperative complications following Clavien–Dindo classification**			
Grade 1 complications, total 44, n			
Ileus Pleural effusion Hepatic marker elevation Nausea/vomiting Skin suture dehiscence	11 9 5 4 4	Renal impairment acute Diarrhoea Chylous ascites Perisplenic fluid collection	2 2 1 1
Grade 2 complications, total 60, n			
Anaemia Skin wound infection Urinary tract infection Pneumonia Sepsis	23 10 7 5 4	Pelvic haematoma Subcutaneous haematoma Pulmonary embolism Anastomotic leak Respiratory insufficiency	3 1 2 1 1
Grade 3 complications, total 7, n			
A Skin burns Pleural effusion	1 1	B Post-laparotomy incisional hernia Anastomotic leak	2 3
**b. Complication related to single patient.**			
**Complications**		**Patients 75, n (%)**	
None		29 (38.6)	
Clavien–Dindo 1		15 (20)	
Clavien–Dindo 2		24 (32)	
Clavien–Dindo 3A		2 (2.5)	
Clavien–Dindo 3B		5.(6.6)	
Clavien–Dindo 4		0	

**Table 5 cancers-14-06243-t005:** General characteristics analysis between no-AL and AL subgroups.

	No-AL ** n 71	AL ** n 4	*p* Value
Age in years, mean ± SD *	61.1 ± 11.3	72.8 ± 6.4	**0.0457**
BMI ^§^ in kg/m^2^, mean ± SD *	25.2 ± 4	21.8 ± 2.9	0.0943
ASA ^¶^, n (%) 1 >1			0.1678
	8 (11.3)	2 (50)	
	63 (88.7)	2 (50)	
Charlson Index, mean ± SD *	7.4 ±1.6	9.5 ±1	**0.0114**
ECOG ^§§^ Performance status n (%) 0 1			0.6148
	36 (50.7)	1 (25)	
	35 (49.3)	3 (75)	
Alcohol n (%)	2 (2.8)	0 (0)	1
Smokers n (%)	7 (9.9)	1 (25)	0.3694
Diabetes n (%)	2 (2.8)	1 (25)	0.1536
Steroid therapy n (%)	1 (1.4)	0 (0)	1
Prior abdominal surgery n (%)	52 (73.2)	2 (50)	0.3113
Prior radiotherapy n (%)	1 (1.4)	0 (0)	1
Prior bevacizumab n (%)	1 (1.4)	0 (0)	1
Preoperative albumin gr/dL, mean ± SD *	33.5 ± 6.3	29.3 ± 10.8	0.2185

* SD: standard deviation; ^§^ BMI: body mass index; ^¶^ ASA: American Society of Anaesthesiologists score; ^§§^ ECOG performance status: Eastern Cooperative Oncology Group performance status; ** AL anastomotic leakage, *p* value in bold type if <0.05.

**Table 6 cancers-14-06243-t006:** Tumour and intraoperative characteristics analysis between no-AL and AL subgroups.

		No-AL ** n 71	AL ** n 4	*p* Value
Ascites n (%)	0 <500 mL 500–1500 mL 1500–3000 mL >3000 mL	38 (53.5) 13 (18.3) 3 (4.2) 10 (14.1) 11 (15.5)	3 (75) 1 (25) 0 (0) 0 (0) 0 (0)	1
Histology n (%)	○ Serous ○ Mucinous ○ Clear-cell ○ Endometrioid ○ Non-epithelial tumour	61 (85.9) 1 (1.4) 2 (2.8) 5 (7) 1 (1.4)	4 (100) 0 (0) 0 (0) 0 (0) 0 (0)	1
Grading n (%)	1 2 3 N/A	0 (0) 5 (7) 69 (97.1) 1 (1.4)	0 (0) 0 (0) 4 (100) 0 (0)	1
Stage n (%)	○ I ○ A ○ B ○ C ○ II ○ A ○ B ○ C ○ III ○ A ○ B ○ C ○ IV ○ A ○ B	0 (0) 0 (0) 0 (0) 0 (0) 2 (2.8) 0 (0) 4 (5.6) 4 (5.6) 45 (63.4) 1 (1.4) 15 (21.2)	0 (0) 0 (0) 0 (0) 0 (0) 0 (0) 0 (0) 0 (0) 0 (0) 3 (75) 0 (0) 1 (25)	1
	Surgical indication n (%) ○ Primary surgery ○ Interval surgery ○ Recurrence surgery ○ Urgent	59 (83.1) 3 (4.2) 8 (11.3) 1 (1.4)	3 (73) 1 (25) 0 (0) 0 (0)	0.2921
	Operating time in minutes, mean ± SD *	333.4 ±81.9	343.3 ±27.5	0.8308
	Intraoperative blood loss, mL, mean ± SD *	414.1 ± 381.2	437.5±415.1	0.9055
	Residual tumour n (%) ○ Absent ○ <1 cm ○ >1 cm ○ Carcinomatosis	48 (67.6) 16 (22.5) 6 (8.5) 1 (1.4)	3 (75) 1 (25) 0 (0) 0 (0)	1
	Intra/postoperative transfusions n (%) ○ None ○ RBC ^§^ ○ plasma ○ platelets	54 (76.1) 15 (21.1) 2 (2.8) 0 (0)	3 (75) 1 (25) 0 (0) 0 (0)	1
	Body temperature at end of procedure °C, mean ± SD *	36.3 ± 0.6	36.7 ± 0.3	0.2638
Resection in cm, mean ± SD *		17.5 ± 6.9	26.0 ± 4.2	**0.0175**
Type of anastomosis n (%) ○ End-to-end ○ End-to-side ○ Pouch ○ Side-to-side Anastomosis procedure n (%) ○ Mechanical ○ Mechanical stapler + reinforcing sutures ○ Manual sutures		59 (80.3) 10 (14.1) 0 (0) 2 (2.8)	2 (75) 1 (25) 0 (0) 0 (0)	0.552
		55 (78.9) 15 (21.7) 1 (1.4)	4 (100) 0 (0) 0 (0)	0.5968
Other intestinal resections n (%) Yes No		7 (9.9) 64 (90.1)	1 (25) 3 (75)	0.3694
Tumour infiltration of intestinal wall n (%) Serosa Muscularis Mucosa		37 (52) 17 (23.9) 17 (23.9)	3 (75) 1 (25) 0	0.8502
Fitness for chemotherapy at 30 days Yes No		61(85.9) 10 (14.1)	1 (25) 3 (75)	**0.0152**

* SD: standard deviation; ** AL anastomotic leakage, ^§^ RBC: red blood cell.

**Table 7 cancers-14-06243-t007:** Published articles on colorectal resection in ovarian cancer surgery.

First Author	Publication, Years	Colorectal Resection, n	AL**, n(%)	Stoma, n (%)	Drains, n %	AL ** Risk Factors
Costantini B et al. [20]	2022	515	15(2.9)	230 (44.7)	NR	BMI ^††^, serum albumin, section IMA ^†††^, medium-low colorectal anastomosis
Lago V et al. [21]	2019	145	7 (4.8)	73 (50.3) 12 colostomy 19 ileostomy 42 ghost ileostomy	NR *	None
Lago V et al. [6]	2019	695	46 (6.6)	267(38.4)	NR *	Age, albumin, additional SBR^#^, manual anastomosis, distance from AV ^§^
Kalogera E et al. [22]	2012	720	43 (6)	0/43	38/43(88.3)	NR *
Kalogera E et al. [5]	2013	126 42 cases 84 controls	42 (33.3)	0 9/84 (10.7)	NR *	Additional LBR ^##^ No protective stoma
Kalogera E et al. [17]	2017	77 309	1(1.7)	27(35.1) 25(8,1)	NR *	NR *
Richardson DL et al. [4]	2006	206	12(5.8)	27 (13.1) 24 end colostomy 3 diverting colostomy	152 (86%)	Serum albumin
Tozzi R et al. [23]	2018	161	5(3.1)	42(26%)	NR *	NR *
Oberaim A et al. [3]	2001	65	3(4.6)	38 (53.8) 33 ileostomy 5 colostomy	NR *	NR *
Grimm C et al. [24]	2017	433	33 (7.6)	NR *	NR *	Albumin serum
Bartl T et al. [25]	2018	169	6 (3.6)	NR *	NR *	MBR ^###^
Yildirim et al. [26]	2014	22	1 (4.5)	NR *	NR *	NR *
Liueca A et al. [27]	2021	40	6 (15)	NR *	NR *	MBR ^###^ partial bowel obstruction, blood loos, PCI^¥^
Bristow RE et al. [28]	2003	31	1(3.2)	0	NR	NR
Mourton SM et al. [29]	2005	70	1(1.4)	20 (17)	NR	NR
Park JY et al. [30]	2006	46	1(2.2)	NR	NR	NR
Peiretti m et al. [31]	2012	238	7(3)	7 (2.8)	NR	NR
Present study	2021	75	4 (5.3)	3 (4)	71 (94.7)	Age, CCI ^†^, length of bowel resection

** AL: anastomosis leakage, * NR: not ^reported §^AV: anal verge, ^#^ SBR: small bowel resection, ^##^ LBR: large bowel resection, ^###^ MBR: multiple bowel resection, ^¥^ PCI: peritoneal carcinomatosis index, ^†^ CCI: Charlson Comorbidity Index, ^††^ BMI: body mass index, ^†††^ IMA: inferior mesenteric artery.

## Data Availability

Data available on request due to privacy restrictions.

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
