# Peer review of "Anastomotic Leakage after Colorectal Surgery in Ovarian Cancer: Drainage, Stoma Utility and Risk Factors"

_cancers, 2022, doi:10.3390/cancers14246243_

Round 1

Reviewer 1 Report

I was asked to review the retrospective manuscript :"Anastomotic leakage after colorectal surgery in ovarian cancer: drainage, stoma utility and risk factors.

The manuscript was interesting, centered on an ever-topical issue concerning rectosigmoid resection during exeresis surgery for ovarian cancer: i.e. the usefulness of the protective stoma and the need for pelvic drainage. The study is well organized even if the number of patients is not as important as that reported in other studies published in the literature

The main objective of the study : i) was to evaluate the anastomotic leakage in patients undergoing intestinal surgery for ovarian cancer in relation to protective stoma and intra abdominal drainage, ii)  identify risk factors of developing anastomotic leakage

As regards the first objective, I have some questions, which the authors should answer: in the material and methods the terms of executions of the colorectal anastomosis are well defined. On the other hand, in the results I find that the flow-chart of the management of patients complicated by anastomotic fistula should be better described.

In particular:

Is CT scanning routinely performed in all operated patients?

When was the CT scan done? If I read correctly, the diagnosis of fistula was made before the drainage showed the presence of fecis?

Once the fistula has been diagnosed, what are the factors that led to re-operating on the patient. It seems to me that every re-operation has not resulted in a stoma but only a drainage. How many fistula patients had a drain? why a surgical and non-radiological drainage?

I think that the small number of patients who have had a fistula does not allow us to answer the second question. Above all, the authors should give an explanation of the fact that the length of the resection is a risk factor. Perhaps these anastomoses are under tension?

Despite these clarifications, the manuscrip deserves to be published

Author Response

Dear reviewer

Thank you very much for the reply and comments on our manuscript. those comment are helpful for improving our work. We have read revisions carefully and we have revised the manuscript according to these suggestions.

Following point to point responses. Relative modifications have been done in the text

“the flow-chart of the management of patients complicated by anastomotic fistula should be better described”

All of the 4 patients (5.3%) who experienced anastomotic fistula the drain content was not the only sign of the complication, as the clinical signs were also highly suggestive. All cases with clinical suspicion of leakage underwent CT scan for confirmation.CT scanning was not performed routinely to all operated patients. Just one patient received conservative treatment, and the others had further surgery with diverting stoma. All patients with AL had drain but no prophylactic intraoperative stoma positioning at primary surgery.   “ the authors should be an explanation of the fact that the length of resection is a risk factor”   A possible explanation that the length of bowel resection is a risk a factor for AL could be the major possibility of the end to end anastomotic tension or devascularization   We hope to have solved requirements.  Looking forward to hearing from you soon we thank you.   Best regards Liliana Mereu

Reviewer 2 Report

Interesting and well-written paper.

The authors analyze the factors that can predict anastomotic leakage in colorectal surgery for ovarian cancer.

They analyzed their case studies and the literature, managing to amalgamate the two researchs into an interesting result.

However, there are some corrections to be made or specifics to be performed.

In the results, in particular in table 4. You have listed all the complications found, but then you have classified them according to Clavien Dindo as if they had occurred in different patients. I suggest making a table listing all the complications and a second table where the authors indicate for each patient what the Calvien-Dindo classification is (if we had two complications for a patient, list only the most serious, it is interesting to know how many patients complicated have, not how many times the individual patient complicated).

in lines 203-204 you write: "it was not possible to continue the treatment program in just 4 patients. " Are the 4 patients the ones who had anastomotic leak? Explains better

In the discussion, you include age as a prognostic factor, explaining that it is a result that you also had in your cohort. In the literature, age is still a debated factor. For example Pirrera et al (Impact of octogenarians on surgical outcome in colorectal cancer. Pirrera B. International Journal of Surgery, 2016, 35, pp. 28–33) do not consider age even greater than 80 years as a prognostic factor of a worse postoperative outcome (including anastomotic leakage). You can cite the article and explain better why age is a factor for you (perhaps the low number of complicated patients may have "altered" the analysis)

Finally in lines 272-274 you write: "Using diverting stoma only in selected patients considerably seems to reduced the incidence and consequent complications reported in other studies on similar populations [6,13,24] without relevant increase of AL complication" It should be explained that the ostomy does NOT REDUCE the incidence of anastomotic complications (which depend on the vascularization and technique) but make the complication less serious due it prevents peritonitis. Explain better in the text.

After these minor corrections, I believe the paper may be ready for publication

Author Response

Dear reviewer

Thank you very much for the reply and comments on our manuscript. those comment are helpful for improving our work. We have read revisions carefully and we have revised the manuscript according to these suggestions.

Following point to point responses

“In the results, in particular in table 4. You have listed all the complications found, but then you have classified them according to Clavien Dindo as if they had occurred in different patients. I suggest making a table listing all the complications and a second table where the authors indicate for each patient what the Calvien-Dindo classification is (if we had two complications for a patient, list only the most serious, it is interesting to know how many patients complicated have, not how many times the individual patient complicated).”

We added table 4b patient-related complications

“in lines 203-204 you write: "it was not possible to continue the treatment program in just 4 patients. " Are the 4 patients the ones who had anastomotic leak? Explains better”

No only 1 patients in the AL group. We explained better in text

“In the discussion, you include age as a prognostic factor, explaining that it is a result that you also had in your cohort. In the literature, age is still a debated factor. For example Pirrera et al (Impact of octogenarians on surgical outcome in colorectal cancer. Pirrera B. International Journal of Surgery, 2016, 35, pp. 28–33) do not consider age even greater than 80 years as a prognostic factor of a worse postoperative outcome (including anastomotic leakage). You can cite the article and explain better why age is a factor for you (perhaps the low number of complicated patients may have "altered" the analysis)”

We agree completely with your comment. We had bibliography and comment in the text

“Finally in lines 272-274 you write: "Using diverting stoma only in selected patients considerably seems to reduced the incidence and consequent complications reported in other studies on similar populations [6,13,24] without relevant increase of AL complication" It should be explained that the ostomy does NOT REDUCE the incidence of anastomotic complications (which depend on the vascularization and technique) but make the complication less serious due it prevents peritonitis. Explain better in the text”

As suggested we explained better in the text

We thank you in advance for take into revision our work We hope to hear you soon.

Best regard 

Sincerely

Liliana Mereu

Round 2

Reviewer 2 Report

Well done.

All suggestions have been collected.

I think the manuscript can be accepted in this form